# New Directions in Diagnostics for Aortic Aneurysms: Biomarkers and Machine Learning

**DOI:** 10.3390/jcm13030818

**Published:** 2024-01-31

**Authors:** Kyle C. Alexander, John S. Ikonomidis, Adam W. Akerman

**Affiliations:** Department of Surgery, Division of Cardiothoracic Surgery, University of North Carolina at Chapel Hill, Chapel Hill, NC 27599, USA; kyle_alexander@med.unc.edu (K.C.A.); john_ikonomidis@med.unc.edu (J.S.I.)

**Keywords:** aortic aneurysm, diagnostic, AI, machine learning, biomarkers, precision medicine

## Abstract

This review article presents an appraisal of pioneering technologies poised to revolutionize the diagnosis and management of aortic aneurysm disease, with a primary focus on the thoracic aorta while encompassing insights into abdominal manifestations. Our comprehensive analysis is rooted in an exhaustive survey of contemporary and historical research, delving into the realms of machine learning (ML) and computer-assisted diagnostics. This overview draws heavily upon relevant studies, including Siemens’ published field report and many peer-reviewed publications. At the core of our survey lies an in-depth examination of ML-driven diagnostic advancements, dissecting an array of algorithmic suites to unveil the foundational concepts anchoring computer-assisted diagnostics and medical image processing. Our review extends to a discussion of circulating biomarkers, synthesizing insights gleaned from our prior research endeavors alongside contemporary studies gathered from the PubMed Central database. We elucidate the prevalent challenges and envisage the potential fusion of AI-guided aortic measurements and sophisticated ML frameworks with the computational analyses of pertinent biomarkers. By framing current scientific insights, we contemplate the transformative prospect of translating fundamental research into practical diagnostic tools. This narrative not only illuminates present strides, but also forecasts promising trajectories in the clinical evaluation and therapeutic management of aortic aneurysm disease.

## 1. Introduction

### 1.1. Background

Aortic aneurysm (AA) disease is intractable. There are many different subtypes which may or may not include a genetic component. The pathology is different based on aneurysm location. Environmental factors, co-morbidities, and sex all differentially affect aneurysm formation and progression. Despite advancements in our understanding of the complex pathobiology of AAs, no efficient method for monitoring exists, and it is becoming clear that no single diagnostic approach will begin to address the many disparate pathological consequences.

The discovery of AA most often occurs during evaluations of unrelated problems. This diagnostic process is inherently sub-optimal, leaving many undiagnosed and at risk for catastrophic complications such as aortic rupture or dissection. Advancements in imaging analysis, biomarker discovery/quantification, and machine learning will form the basis for improved monitoring of patients with AAs.

### 1.2. Significance

An AA is a localized, progressive dilatation of the aorta that, if unidentified, can result in catastrophic outcomes. Pathogenesis covers a wide range of idiopathic–degenerative, congenitally acquired, genetically based, and traumatically induced disorders of the aorta [1]. Each year in the United States, approximately 10,000 people die from AAs, while over 16,000 people die from associated complications, making AA the 18th most common cause of death in the US, and the 15th most common cause of death in individuals older than 65 years of age. However, given the predominantly asymptomatic nature of AA, the incidence can be difficult to estimate and may be much higher than originally expected.

If a patient is fortunate enough to be diagnosed early on, a “watch and wait” surveillance program is initiated until the risk of aortic rupture outweighs the risk of surgical repair. Aneurysm occurs in both the thoracic (TAA) and abdominal (AAA) regions of the aorta, the former accounting for 25% of cases, the latter, 75% [2]. As such, regional differences in the etiology, incidence, and diagnosis of aortic disease in the thoracic versus abdominal aorta should be carefully considered. 

Regardless of location, enhanced proteolysis results in pathological tissue remodeling and progressive dilation. Analysis of the natural history of AAs revealed that aneurysms progress at a continuous rate, i.e., 0.1–0.3 cm per year in the thoracic aorta [3]. Moreover, this analysis also identifies sharp “hinge points”. In the thoracic region, for example, life-threatening complications are all but guaranteed above 6 cm [3]. Thus, early identification will mitigate associated morbidity and mortality.

While aortic disease is more prevalent in men, the prognosis is typically worse in women who, at smaller aortic diameters, are more likely to suffer an aortic dissection or experience rupture [4]. Aortic pathology is often overlooked and diagnosed later in women; therefore, they experience higher rates of other associated harms, such as major surgical complications and hospital readmission [5].

Autopsy studies have revealed that the most common cause of death due to AA is rupture [6]. This highlights the critical need for developing a standardized screening technique for early diagnosis to mitigate life-threatening complications. While it is likely that the degenerative process begins earlier in life, the mean age at diagnosis is approximately 65 years [7]. Previous reports have demonstrated that the prevalence of AA doubled between 1982 and 2002 [8,9,10]. Population projections suggest that, as the “Baby Boomer” generation ages, the number of individuals 65 years and older will double by the year 2030 [11,12]. The number of patients living with AA is sure to rise [13]. 

In addition to high morbidity and mortality, the treatment of advanced AA disease is especially resource-intensive. In a recent report published in the *Journal of the American Heart Association*, an analysis of the economic burden of AA identified an increasingly substantial capital commitment by the healthcare sector for a relatively small percentage of the population [14]. Consequently, AAs represent a disproportionate share of the burden on hospital resources and expenses; early identification would mitigate this.

At present, diagnosis is dependent on costly advanced imaging techniques, such as computed tomography (CT) and magnetic resonance imaging (MRI). There are no point-of-care blood tests available that screen for or follow AA progression to inform optimal timing for surgical intervention. Indeed, The US Preventive Serves Task Force (USPSTF) recommends screening for infrarenal abdominal aneurysms with ultrasonography in men aged 65 to 75 years who have ever smoked [15]. In this case, the necessity for high-throughput diagnostics is clear: conducting such a large number of ultrasounds will strain, critically, primary care providers. Importantly, diagnoses of abdominal AA are missed by ultrasound in 30% of patients [16]. While ultrasonography has been successful in identifying larger aneurysms in the infrarenal abdominal aorta, as a means of early detection it is unsuccessful [17]. Additionally, the aortic arch and descending thoracic aorta are far more difficult to image using surface ultrasonography due to interference and shadowing from the nearby ribs and air-filled lungs [18]. Ultrasonography is operator-dependent, and it often underestimates the aortic diameter by 2–5 mm [19,20]. The fact that there is an urgent need for better, more accurate diagnostic tools is alarmingly clear.

## 2. Computer-Assisted Measurement of the Aorta

### 2.1. Artificial Intelligence and Machine Learning

Diagnostic CT imaging of the aorta is of paramount importance, with centerline measurements being the gold standard, because imaging is the only way to detect aortic dilation and cross-sectional imaging is the only means of depicting the entirety of the aortic arch [21]. Measurement accuracy can be problematic, however. Elefteriades et al. have asserted that 1–2 mm is insufficient to detect change, and it is unlikely that a measured change of 3–4 mm is a sufficient diagnostic criterion [13]. Pradella et al. were in agreement when they investigated inter-observer variability [21]. Ultimately, the two teams concluded that radiologists’ measurements of aortic diameters from chest CT scans usually involve impreciseness of up to 3–4 mm [13,21]. It is clear that inter-observer variability must be minimized to provide better patient care.

Siemens’ artificial intelligence (AI)-Rad Companion (Siemens Healthineers, Erlangen, Germany) is a decision support tool for the radiological assessment of CT images of the thorax. The AI-Rad Companion combines AI models and machine learning paradigms, namely, so-called ‘deep’ learning, reinforcement learning, convolutional and generative adversarial networks and symmetric convolutional encoder–decoder architecture, image-to-image masking, and supervised machine learning models. (Written Report WR, Siemens Healthineers, Erlangen, Germany, Brader, May 2023, “AI-Rad Companion Chest CT” at Diagnostikum Linz).

Siemens’ algorithm has been trained on over 1250 CT datasets, including native and contrast-enhanced scans. The aorta analysis pipeline consists of landmark detection, aorta segmentation, and diameter measurements (WR). The algorithm automatically detects six aortic landmarks, as per AHA guidelines (Aortic Root, Aortic Arch Center, Brachiocephalic Artery Bifurcation, Left Common Carotid) [22]. Accurate measurement of the aorta in accordance with the guidelines is essential for reporting comprehensive aneurysm expansion, as this information can impact the choice of interventional repair strategy. The assistance of AI holds the potential to enhance reporting efficiency and significantly decrease inter-reader variabilities among radiologists, thereby improving the accuracy of diagnostic follow-up [23]. 

Deep learning, AI, and ML algorithms represent immensely valuable ways to minimize inter-observer variability and save time. Pradella et al. reported no variance in repeated deep learning measurements of the same case, constituting perfect preciseness [21]. While centerline-based measurements have been the preferred method for nearly two decades, evaluation of thoracic aorta dimensions through measurements perpendicular to the vessel’s centerline remains a time-intensive process, taking about 5–6 min per case [21]. Marschner et al. reviewed the deep image-to-image network (DI2IN) algorithm from Siemens, which automatically generates contours without user input, and takes roughly 30 s [24]. AI assistance can reduce the time it takes to examine and annotate radiological images. 

AI-Rad Companion Chest CT correctly assessed the presence or absence of thoracic aortic dilatation in 17,691 exams (97%) out of 18,243, including 452 cases with previously missed dilation independent from contrast protocols [21]. These findings suggest its usefulness as a secondary reading tool that will improve report quality and efficiency.

The average reading time was also the focus of a study by Rueckel et al., which included patients with aortic ectasia undergoing follow-up assessments. The average reading time was reduced by 63%. Moreover, AI assistance reduced total diameter inter-reader variability by 42.5% [23]. In a study by Yacoub et al., chest CT reading times from three radiologists were evaluated (N = 390). The mean reading time with AI assistance was reduced by 22.1% [25]. The average absolute error in aorta diameters was 1.6 mm across all nine measurement locations and varied between 1.2 mm and 2.2 mm per location (N = 193) (WR).

Siemens’ AI-Rad Companion is limited currently to the chest and cannot analyze the abdomen or other body regions. While other open-source software programs do have the potential to be leveraged in other areas of the body, at present, they do not have Siemens’ built-in guideline compliance and users must have the baseline technical facility to navigate programming languages [23].

In addition to Siemens AI-Rad companion, the open-source PyRadiomics platform, implemented in the Python programming language, can provide similar information assessment and workflows. PyRadiomics works with other open-source, Python-based pipelines, such as the Insight Toolkit, SimpleITK, PyWavelets, and NumPy and in conjunction with another open-source technology, 3D Slicer. The PyRadiomics platform extracts radiomic data from medical imaging modalities such as CT, PET, and MRI, following four main steps: loading and preprocessing images and segmentation maps, applications of enabled filters, calculations of features using different feature classes, and returning results.

Loading and preprocessing of medical images, along with the management of segmentation maps, can be carried out using SimpleITK, an open-source Insight Toolkit. To ensure uniform distances between isotropic and neighboring voxels in all directions for texture and shape features, various resampling options are available. Filtering can be applied directly to the image or through built-in options, including wavelet and Laplacian of Gaussian (LoG) filters, as well as simple filters such as square, square root, logarithm, and exponential filters. PyWavelets and SimpleITK are employed for the wavelet and LoG filter, while *NumPy* is used for the remaining filters.

Feature calculation encompasses five feature classes: first-order statistics, shape descriptors, and texture classes such as the gray-level concurrence matrix, gray-level run length matrix, and gray-level size zone matrix [26,27]. All statistical and texture classes support feature extraction from both filtered and unfiltered images. Feature extraction is applicable to both single-slice (2D) and whole-volume (3D) segmentations. The pyradiomics script handles single-image processing, while the pyradiomics batch script handles batch processing. 

### 2.2. Machine Learning Concepts

This section presents a short list of ML concepts that are commonly used in image processing, computer vision, and biological computing. 

Within the context of machine learning, ‘adversarial’ refers to training two models concurrently by having them play out a zero-sum game in which a ‘generator’ and ‘discriminator’ compete against one another: the generator generates images in an effort to fool the discriminator; the discriminator compares the generated images with real ones defined beforehand as ground truth in order to rightfully discriminate between them [28].

Convolutional neural networks (CNNs) are commonly used for image-to-image processing tasks due to their ability to learn hierarchical features from images. The encoder extracts features from the input image, and the decoder generates the output image based on these features. Spatial information is preserved with convolutional matrices, skip connections, and weighted values determined during the adversarial, zero-sum game played by the models and the training process in which pairs of images are fed into the network, allowing it to learn to minimize the difference between its predictions and the ground truth defined by radiologists [29].

Image-to-image processing in the context of CNNs or generative adversarial networks (GANs) involves tasks where the goal is to transform an input image into an output image. These tasks include various image translation or transformation problems, such as style transfer, image segmentation, super-resolution, etc. Convolutional encoder–decoder refers to a neural network architecture commonly used in image processing tasks. The encoder part extracts features from the input data through convolutional layers; the decoder part reconstructs the output from these features using up-sampling layers [30].

In this context, feature concatenation refers to the combining or concatenating of features extracted by the encoder before passing them to the decoder. This can enhance the information available for the decoding process. Deep supervision networks involve incorporating supervision signals (e.g., intermediate layer outputs) at multiple levels of the network. This can help improve gradient flows during training and potentially improve convergence. Multi-level blocks indicate that the architecture consists of multiple hierarchical levels or blocks, possibly with different levels of abstraction. These elements are commonly used in advanced image processing and analysis tasks to preserve image features and fidelity and facilitate algorithmic analyses of complex radiomic data [31].

### 2.3. Considerations Regarding AI Limitations

AI and ML are, of course, not without certain limitations. So-called ‘black box’ limitations refer to ML model predictions that generate results which cannot be explained or understood [32]. The innerworkings of the model are poorly understood; therefore, interpreting the reasons for the model’s output is challenging. This limitation is important for several reasons. Traditional statistical models operate on interpretable rules and coefficients that allow users to understand how variables contribute to calculated results; however, in black-box models, the relationships between inputs and outputs are not easily understood, making the model’s decision-making process incomprehensible. Moreover, it can be difficult to trust a model’s predictions, especially in the healthcare sector, if its reasoning cannot be followed; this lack of transparency also raises concerns about potential errors and biased or incorrect variable interpretation, and these issues cannot be addressed if the input errors cannot be identified [32]. While efforts are being made to address these issues, especially in the field of explainable AI (XAI), there is still much work to be done.

Despite these limitations, however, these findings demonstrate that ML algorithmic image processing, segmentation, and analysis are effective and functional aids for researchers and clinicians and can significantly improve diagnostic workflows, supplementing a more holistic approach. Importantly, ML can also be used to analyze and incorporate the data it generates (or any other tabular datasets) to achieve better personalized medicine generally. Of course, ML algorithms are not meant to replace trained radiologists, but they can be valuable tools to aid them. Finally, biochemical monitoring can work in concert with ML to build better diagnostic tests and criteria.

## 3. Biochemical Monitoring

### 3.1. Background

Numerous studies have explored viable alternatives to imaging for detecting aneurysm disease. Several methods have included analyzing various “pathology” indicators such as circulating immune cells [33,34], markers of inflammation [35,36,37], hemostasis [38], acute-phase proteins [39,40], and plasma homocysteine levels [41,42,43]. Unfortunately, the presence of these analytes in the bloodstream can also result from recent surgery and other disease processes, problematizing diagnostic predictions.

Investigations have largely focused on the discovery of biomarkers for abdominal AAs. An outstanding review recently published in the *Journal of Clinical Medicine* provides an excellent summary of 25 studies that have identified specific “clinically applicable” and “experimental” biomarkers for AAs [44]. However, conclusions drawn from this retrospective analysis were somewhat anticipated: “The current literature provides a plethora of data with conflicting results and firm conclusions cannot be provided”. Given the existing hurdles in using biomarkers to predict aneurysm expansion in clinical settings—such as their lack of disease specificity and inability to cover all types of AA—an integrated prognostic model that combines select circulating markers will offer enhanced clinical utility [45]. Nevertheless, results consistently demonstrate that circulating biomarkers can be used to identify aneurysms and form the basis of an individualized surveillance strategy to discern risk. Although it is evident that these biomarkers signal the progression of pathology, creating a clinical assay based on them has proven to be challenging.

### 3.2. Genomic/Proteomic Analysis

In 2007, Wang et al. proposed a 41-panel gene signature array to identify the presence of TAAs [46]. This study successfully demonstrated that gene expression patterns in circulating leukocytes could predict the status and subtype of TAAs. More recently, Marshall et al. found elevated levels of fibrillin fragments in aneurysm patients, with concentrations varying in different anatomical locations of aneurysm (thoracic vs. abdominal) [47]. Despite the strengths of these studies, the authors were unable to definitively establish their uses as monitoring or diagnostic techniques for AAs.

In a separate investigation, proteomic analysis identified several markers for thoracic aortic aneurysms [48]. Among these, the four and a half LIM domain protein 1 (FHL1) emerged as the most useful in predicting TAA. FHL-1 was combined with Collagens I, III, V, and XI in a five-panel marker test, where upregulation of any three by over 50% successfully identified the presence of TAAs. Although these studies have made significant contributions, a successful method for biochemically monitoring aortic aneurysm disease remains obscure. Furthermore, the methodologies used in these past investigations may be too complex to expand effectively. Implementing quantitative and scalable approaches is essential for their practicality in a clinical setting.

### 3.3. Circulating Protein Quantification

Phase I results from the National Registry of Genetically Triggered Thoracic Aortic Aneurysms and Cardiovascular Conditions (GenTAC) trial revealed that circulating levels of transforming growth factor beta (TGF-β) are increased in Marfan (MFS) patients with thoracic AAs [49]. Specifically, the authors demonstrated that circulating TGF-β1 concentrations are elevated in MFS and decrease after the administration of losartan, beta-blocker therapy, or both, and therefore might serve as a prognostic and therapeutic marker in MFS patients with TAA. Given its pivotal role in vascular pathology and the maintenance of the extracellular matrix, there is considerable interest in investigating the impact of TGF-β1 on vascular remodeling. Its immense promise as a biological indicator for tracking pathology progression further accentuates this interest.

Both intracellular and extracellular mechanisms function to balance matrix deposition and degradation to maintain structural integrity of the aortic wall. In AAs, this balance becomes disrupted in favor of enhanced proteolysis, resulting in pathological remodeling, and leading to progressive dilation. Vascular remodeling is an important process in which a critical family of proteolytic enzymes, the matrix metalloproteinases (MMPs), actively participate through degradation of the vessel wall and the subsequent release of sequestered growth factors and cytokines, such as TGF-β [50,51]. This breakdown of normally long-lasting matrix molecules, such as elastin and collagen, has placed a great deal of emphasis on the importance of research focusing on the involvement of MMPs in AA. Multiple studies have demonstrated differential expression profiles of MMPs and their endogenous inhibitors, the tissue inhibitors of MMPs (TIMPs) in clinical AA specimens and animal models.

It has been shown that AAs can be identified in plasma by profiling the MMP/TIMP ratio as it provides a unique metric of aortic wall remodeling [50,51]. These proteolytic enzymes degrade all components of the vessel wall and are attributed to the development and progression of AA [52,53]. Alterations in the MMP/TIMP ratio may also be indicative of AA presence, location, and severity.

The extracellular MMP inducer (EMMPRIN), also called CD147, is a cell surface transmembrane glycoprotein that, mainly through interacting with cyclophilin A, is involved in several cellular processes including the induction of MMPs and the migration, inflammation, and transport of nutrients [54]. MMPs are known to facilitate pathological remodeling and EMMPRIN is directly involved in MMP production; thus, it is likely that EMMPRIN plays an important role in pathology. EMMPRIN is secreted from vascular smooth muscle cells in AA [55] and its expression is induced by angiotensin II and TGF-β administration in vitro [55,56].

A study of MFS patients with aortic ectasia found that EMMPRIN levels were markedly reduced; when compared with healthy controls, this proved predictive of ectasia [57]. This study attested that monitoring circulating EMMPRIN, in combination with current diagnostic tools, can effectively track aortic diameters. Importantly, circulating levels of the aforementioned proteins (MMPs, TIMPs, EMMPRIN, and TGF-βs) are all quantifiable using the high-throughput, immune-based, multiplexed screening platform: Multiplex Suspension Array [58].

### 3.4. Circulating microRNA Quantification

MicroRNAs, a class of small, non-coding RNA, function to regulate translation by interaction with the 3′-untranslated region (UTR) of targeted mRNAs [59]. Increasing evidence supports a direct role for altered microRNA abundance in pathological cardiovascular remodeling and disease progression. Alterations in microRNA abundance are emerging as a clear mechanism mediating changes in matrix remodeling pathways associated with AA. However, measuring microRNAs in blood poses challenges, primarily due to the absence of consistent, widely accepted protocols. However, by instituting standardized procedures and robust quality control measures, researchers and clinicians can heighten the dependability and precision in measuring circulating microRNAs [60]. This endeavor will bolster swift progress in molecular diagnostics and personalized medicine fields.

Multiple microRNAs are endogenous upstream regulators of many key proteins involved in aneurysm progression, and have been demonstrated to directly regulate cellular phenotype and extracellular matrix remodeling [61,62,63]. In combination with MMP and TIMP concentrations, microRNA levels, when united with multivariable stepwise regression, have shown significant promise in the algorithmic detection of thoracic AAs: these can identify and distinguish between etiological subtypes of TAA with an accuracy exceeding 95% [58]. Moreover, a linear correlation exists between circulating levels of several of these microRNAs and aortic diameters, suggesting that quantification may be used as a predictor of risk [58,61].

### 3.5. Circulating Extracellular Vesicle Concentration, Size Distribution, and Cargo Analysis

More circulating targets for AA diagnosis and monitoring are emerging. Wang and colleagues demonstrated the involvement of extracellular vesicles (EVs) derived from macrophages in the pathogenesis of abdominal aortic aneurysms [64]. EVs play a crucial role in cell-to-cell communication, comprising exosomes (sized between 30 and 100 nm) and microvesicles (ranging from 100 to 300 nm), originating from various cell types. Studies highlight that circulating EVs harbor MMPs, TIMPs, and microRNAs. Furthermore, vascular cells release multiple microRNAs in EVs due to disease progression [65]. This results in altered circulating microRNA profiles, reflecting the origin and location of an aneurysm, thereby establishing distinct EV contents specific to different types of aortic aneurysms. In addition to alterations in cargo, EV concentrations and size distributions are altered in MFS patients with TAA, suggesting that profiling them will establish their clinical utility as a novel diagnostic [66].

These findings illustrate EVs’ diagnostic potential, but EVs are technically difficult to examine and analyze; therefore, isolation protocols must be optimized and standardized to ensure consistent quantification and the accuracy of size-based categorization. Similarly to assessments of earlier protein and nucleic acid targets, quantifying EVs requires implementing standardized procedures and robust quality control measures [66]. This approach aims to enhance the reliability and precision of measurements and subsequent downstream analyses for researchers and clinicians.

## 4. Conclusions

In pursuit of an innovative frontier in diagnostic modalities, our focus gravitates towards developing computational tools primed to both generate and meticulously analyze data. This endeavor takes shape through the comparison of aortic diameter measurements procured from AI-aided analysis of CT scans, amalgamated with patient-specific data and biochemical profiles. We intend to harness the potential of supervised and unsupervised ML methodologies to manipulate, categorize, and visually represent data. The overarching objective remains the establishment of a standardized screening technique, a pivotal step in the identification of aneurysms. This concerted effort aims not only to stratify risk, but also to mitigate life-threatening aortic complications within the broader population.

An enhanced biochemical monitoring tool designed as a simple blood test holds immense clinical significance on several fronts. Chiefly, it provides a non-invasive and easily accessible means of routinely monitoring patients, allowing for frequent assessments without subjecting individuals to invasive procedures or advanced imaging techniques. This translates to the more frequent and timely monitoring of aortic aneurysm disease progression or changes in biochemical markers associated with the condition.

Numerous circulating biomarkers associated with aneurysm disease have been recognized, potentially improving treatment decision making and optimizing precision medicine. However, their quantitative measurement is crucial for diagnostic purposes. Moreover, larger prospective trials are needed to establish and evaluate prognostic models that offer the greatest benefit to the general population.

These tools can significantly improve early detection and risk stratification. By regularly measuring imaging data and specific biomarkers associated with aortic aneurysm disease, clinicians can identify patients at higher risk of complications in the disease process. Early detection could prompt timely interventions or closer monitoring for those at higher risk.

Imaging is an invaluable staple modality that will likely always be necessary. AI implementation will make imaging better and more efficient. Combining ML strategies with biochemical monitoring will improve diagnostic efficacy. Moreover, the simplicity and accessibility of better diagnostics for the monitoring of aortic aneurysm disease will have broader implications for healthcare equity. Simpler, more accessible diagnostics can better serve marginalized communities, improve overall patient care, and democratize existing disparities in diverse populations.

## Data Availability

Not applicable.

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
