# Peer review of "New Directions in Diagnostics for Aortic Aneurysms: Biomarkers and Machine Learning"

_jcm, 2024, doi:10.3390/jcm13030818_

Round 1

Reviewer 1 Report

Comments and Suggestions for Authors

The authors describe aspects of future developments in the fields of artificial intelligence, machine learning and the analysis of various biomarkers for predicting development or progression of aortic aneurysms. However, the informations given should be more focused on the field of aortic aneurysmal disease discussing the publications and the value of the data published. The authors should focus on the field of their own competence. 

1) The remarks about artificial intelligence and machine learning is more a summary of different computer techniques that might play a role in the future without giving real profound data or showing the current status of these techniques.

2) The information given about biomarkers are not new, most references are from 2010 are even older; most of these biomarkers have not shown any important role within the last 10 years. 

3) Probably of interest would be the field about the circulating extracellular vesicle levels, a topic in which the authors have contributed experimental data published recently.  

Therefore, it would be more interesting to provide a condensed review about the most recent development regarding new new biomarkers having shown a close correlation to disease aneurysm diameter, progression or indicate new therapeutic or experimental options for treatment.  

Comments on the Quality of English Language

the language should bei clear and briefly describing the scientific work  

Author Response

We would like to thank this reviewer for taking the time to carefully evaluate this manuscript. Below is a point-by-point response to all comments.

In response to Reviewer 1’s comments on AI and ML, we would like to highlight that Lines 97-175 meticulously detail the current clinical applications of AI. Notably, Siemens' AI-Rad companion is approved for clinical use in the United States and European Union, with the cardiovascular package having been approved in the US in late 2023. A summary of the data which shows the current status of AI and ML in the context of aortic aneurysm disease is located in lines 114-144 and in citations 21 and 23-26.

Regarding Reviewer 1’s concerns about biomarkers: We appreciate the acknowledgment of historical references. This was the overall point of the article. The fact that no biomarker has been used successfully as a diagnostic in over a decade is precisely why we are suggesting the incorporation of AI and ML modeling with biomarker panels and patient information. This is known as applied science. Regarding extracellular vesicles, this was a small portion of the topic that we were asked to write about and there is limited information currently available in the context of diagnosing aortic aneurysm disease. Therefore, as more studies are published, we will gladly make that the topic of our next review.

In addressing Reviewer 1’s comments on author competence: We would like to emphasize the substantial contributions of the authors. Adam Akerman, an Assistant Professor and director of the UNC Cardiovascular Research Laboratory, has established a state-of-the-art translational research facility, small animal surgical operatory, and large clinical tissue and plasma repository for the study of aortic aneurysm disease. With a BS in biology, an MS in microbiology and immunology, and a PhD in molecular cellular biology and pathobiology, Dr. Akerman is an established investigator, funded by the NIH/NHLBI, with over fifteen years of experience studying aortic aneurysm disease, biomarker discovery, assay development, biochemical analysis, cellular and molecular biology, microRNAs, and extracellular matrix remodeling.

      John Ikonomidis, a cardiothoracic surgeon, and Chief of the Division of Cardiothoracic Surgery at UNC Health. He performs numerous aortic procedures annually and maintains a productive basic science/translational research laboratory. As a principal investigator, Dr. Ikonomidis has retained continuous NIH support for more than two decades. He has served as chair of the Bioengineering, Technology and Surgical Sciences (BTSS) Study Section at the NIH and is on the Data and Safety Monitoring Board of the Cardiothoracic Surgical Trials Network. Dr. Ikonomidis also serves on the editorial boards of the Journal of Heart Valve Disease, The Journal of Cardiac Surgery, Circulation, Aorta, and The Journal of the American Heart Association, and is an associate editor of the Journal of Thoracic and Cardiovascular Surgery. Throughout his longstanding career as an eminent physician scientist, Dr. Ikonomidis has conducted many foundational aortic studies.

      Kyle Alexander is an expert in science writing and editing who has extensive experience in information science and machine learning. Concerning the quality of English language usage: we assure the reviewer that the manuscript benefits from the expertise of Kyle Alexander. Holding a B.A. in English from The Citadel, Cum Laude, an M.A. in British Literature post-1660, Summa Cum Laude, from the University of South Carolina, and being A.B.D. Summa Cum Laude in Eighteenth-Century British Literature, he brings over 8 years of experience teaching writing, composition, rhetoric, and writing style at the college level. Therefore, we believe it is reasonable to conclude that the topic of this invited expert review is well within the “field of [our] own competence”.

Reviewer 2 Report

Comments and Suggestions for Authors

This review article presents a thorough examination and evaluation of groundbreaking technologies that are poised to revolutionise the diagnosis and management of aortic aneurysm disease. The comprehensive analysis provided is based on an exhaustive review of both contemporary and historical research, with particular emphasis on the integration of machine learning (ML) and computer-aided diagnosis.

The review draws extensively on relevant work, including insights from Siemens' published testimonial and numerous peer-reviewed publications. At the core of our review is an in-depth examination of ML-driven diagnostic advances, dissecting various algorithmic suites to reveal the fundamental concepts that underpin computer-aided diagnostics and medical imaging.

The review extends its reach to discuss circulating biomarkers, synthesising findings from previous research efforts and contemporary studies sourced from the PubMed Central database. The authors highlight prevailing challenges in the field and envision the potential fusion of artificial intelligence (AI)-guided aortic measurements and sophisticated ML frameworks with computational analyses of relevant biomarkers.

In conclusion, this narrative not only makes a significant contribution to the existing body of knowledge in aortic aneurysm research, but also serves as a valuable resource for clinicians, researchers and policy makers alike. The foresight of the authors lays the foundation for future innovations in the clinical evaluation and therapeutic management of aortic aneurysm disease. The meticulous attention to detail, coupled with a forward-looking perspective, makes this review an essential part of the evolving landscape of cardiovascular medicine.

Summarizing, I see potential of this submitted manuscript.

Author Response

We would like to thank reviewer 2 for their time and gracious comments.

Round 2

Reviewer 1 Report

Comments and Suggestions for Authors

The authors have improved some points in their manuscript; which is an enthusiastic presentation about the role of AI, ML and a mixture of old and few new biomarkers. 

Please make clear that the summary combines several ideas which are still rather speculative and only supported by few real data. 

In the abstract please remove the sentence from line 11 to line 15, it does not affect the message of the abstract.

In line 373, page 8, please include the word "might" instead of "can".  

Comments on the Quality of English Language

minor corrections    

Author Response

Reviewer: The authors have improved some points in their manuscript; which is an enthusiastic presentation about the role of AI, ML and a mixture of old and few new biomarkers.

Response: We are grateful for both the reviewer’s assessment in this case and their acknowledgement of our enthusiasm for this manuscript’s subject matter.

Reviewer: Please make clear that the summary combines several ideas which are still rather speculative and only supported by few real data.

Response: Again, we are thankful for this reviewer’s interest and careful consideration, but as they don’t specify any discrete ideas, elaborate on what constitutes speculation, or provide constructive ideas or advice about which data (or how that data might support an idea), it is hard to address this concern in meaningful ways.

Of the ideas in the manuscript, it is difficult to describe any of them as ‘speculative.’

AA disease, its incidence, pathogenesis, significance, etiology, diagnosis, prognosis, consequences, et cetera, is well known and well described.

Medical imaging and Computer Assisted Diagnosis (CAD) are not speculative.

AI and ML are established, ubiquitous technologies: while there are still challenges to be overcome regarding their design and implementation in medicine, neither their capabilities nor the mathematical/statistical models upon which they are built are speculative. Copious data illustrating the above are included in the manuscript and the citations.

The Siemens AI-Rad Companion and the SciPy suite are novel technologies, but their efficacy is well established, and utility is varied and widespread.

Reviewer: In the abstract please remove the sentence from line 11 to line 15, it does not affect the message of the abstract.

Response: “This overview draws heavily upon relevant works, including Siemens’ published field report and many peer-reviewed publications. At the core of our survey lies an in-depth examination of ML-driven diagnostic advancements, dissecting an array of algorithmic suites to unveil the foundational concepts anchoring computer-assisted diagnostics and medical image processing.” (lines 11-15)

If the quoted text from lines 11-15 “does not affect the message of the abstract”, leaving it unaltered should be equivalent to its removal, given that the text may be judged subjectively to be irrelevant. As such, we advocate for its remaining in the text as it is. Further, as these sentences do, in fact, contextualize broadly the trajectory of the review and one of its primary purposes, we are not inclined to remove it. Should academic editors also desire its removal, we are happy to oblige, however.

Reviewer: In line 373, page 8, please include the word "might" instead of "can".

Response: The text around the word “can” in line 373 reads: “These tools can significantly improve early detection and risk stratification. By regularly measuring imaging data and specific biomarkers associated with aortic aneurysm disease, clinicians can identify patients at higher risk of complications in the disease process. Early detection could prompt timely interventions or closer monitoring for those at higher risk.”

This point seems self-evident, insofar as regular analysis of image and biomarker data enables clinicians to assess accurately and sensitively a patient’s condition over time. “Can” as a transitive verb denotes “expressing physical or mental ability: be able to, know how to; have the power, ability, or capacity to” (OED). The word choice is connotatively and denotatively appropriate in this context. Therefore, weakening the statement by substituting the conjectural/conditional ‘might’ seems ill advised in this case, as clinicians’ ability to perform or do this isn’t in doubt or speculative.